# Effect of Frequency Coupling on Stability Analysis of a Grid-Connected Modular Multilevel Converter System

**Yixing Wang** [1], **Qianming Xu** [1,*] **and Josep M. Guerrero** [2]

1   College of Electrical and Information Engineering, Hunan University, Changsha 410000, China; wang1809@hnu.edu.cn
2   Department of Energy Technology, Aalborg University, 9200 Aalborg, Denmark; joz@et.aau.dk
*   Correspondence: xqm@hnu.edu.cn; Tel.: +86-13677375126

**Abstract:** Due to the internal dynamics of the modular multilevel converter (MMC), the coupling between the positive and negative sequences in impedance, which is defined as frequency coupling, inherently exists in MMC. Ignoring the frequency coupling of the MMC impedance model may lead to inaccurate stability assessment, and thus the multi-input multi-output (MIMO) impedance model has been developed to consider the frequency coupling effect. However, the generalized Nyquist criterion (GNC), which is used for the stability analysis of an MIMO model, is more complicated than the stability analysis method applied on single-input-single-output (SISO) models. Meanwhile, it is not always the case that the SISO model fails in the stability assessment. Therefore, the conditions when the MIMO impedance model needs to be considered in the stability analysis of an MMC system should be analyzed. This paper quantitatively analyzes the effect of frequency coupling on the stability analysis of grid-connected MMC, and clarifies the frequency range and grid conditions that the coupling effect required to be considered in the stability analysis. Based on the quantitative relations between the frequency coupling and the stability analysis of the grid-connected MMC system, a simple and accurate stability analysis method for the grid-connected MMC system is proposed, where the MIMO impedance model is applied when the frequency coupling has a significant effect and the SISO impedance model is used if the frequency coupling is insignificant.

**Keywords:** modular multilevel converters (MMC); impedance; frequency coupling; stability



## 1. Introduction

Due to the modularity, scalability, high efficiency, and superior harmonic performance [1–3], modular multilevel converters (MMCs) are increasingly used in high-voltage/high-power applications [4,5], e.g., high-voltage direct current (HVDC) transmission and static synchronous compensation (STATCOM). However, the resonant interactions among the MMCs and electrical systems may result in harmonic instability in these systems [6]. When MMC is connected to a weak grid, the stability margin of the system is significantly reduced, which can easily lead to resonance problems [7,8]. The instability issues in the MMC system have been reported in [9,10]. It is essential that the stability of the grid-connected MMC system is considered.

The impedance-based analysis method is an effective method for assessing the system stability [11], and admittance/impedance modeling is a prerequisite to applying the analysis [12,13]. Since the MMC has become the state-of-the-art technology for high-power applications, the admittance/impedance modeling of MMC has attracted extensive research. The impedance model of an MMC connected to a wind farm is developed in [14], where the internal harmonics of an MMC are included based on the harmonic state-space (HSS) modeling method. The impedance model of a grid-connected MMC is derived in [15,16], where the impedance shaping effect produced by different control schemes was fully elaborated. In the above studies, the admittance/impedance of MMC is seen as the single-input single-output (SISO) form. When the symmetric control is adopted in the

two-level voltage source converter (VSC), the decoupled SISO admittance/impedance is considered reasonable in the VSC [17,18]. However, different from the VSC, the coupling between the positive and negative sequences in impedance, i.e., frequency coupling, exists in the MMC even under the symmetric control. As concluded in [19,20], the frequency coupling has an impact on the stability assessment of grid-connected VSC at low frequency. Thus, the frequency coupling of MMC admittance should be considered in the stability assessment of a grid-connected MMC system.

Recently, the frequency coupling of MMC admittance/impedance has received considerable attention [21–29]. Two-order MMC impedance matrixes are proposed in [21–26], where the off-diagonal elements in the matrix capture the frequency coupling in MMC impedance. A three-order impedance model of MMC is derived in [27], and the impedance model includes the coupling among the positive-sequence system, the negative-sequence system, and the DC system. The authors of [28,29] proposed an equivalent SISO impedance model of grid-connected MMC considering its frequency coupling, and the model shows that the frequency coupling makes the MMC impedance be coupled with the grid impedance.

Previous research has shown that ignoring the frequency coupling may lead to inaccurate stability assessment of the interconnection systems [20,21]. Although the SISO impedance model provides simplicity and convenience for the stability analysis, some cases exist where the SISO impedance model leads to the wrong stability assessment [30]. To include the frequency coupling effect, [21–27] apply the MIMO impedance model of MMC in the stability analysis of MMC-HVDC, where the generalized Nyquist criterion (GNC) is inevitably introduced in the stability assessment. However, the stability assessment applied by the GNC is more complicated than the SISO analysis tool. Meanwhile, it is not always the case that the SISO model fails in the stability assessment. Therefore, the condition that the frequency coupling must be considered in the stability analysis of grid-connected MMC deserves to be studied.

This paper quantitatively analyzes the effect of frequency coupling on the stability analysis of grid-connected MMC. Based on the amplitude comparison between the sequence impedance and coupling term of the MMC under different control schemes, the frequency range that the coupling should be considered in the stability analysis is given. Furthermore, following the stability assessment procedure of the grid-connected MMC system, the grid conditions when the coupling effect needs to be considered in the stability analysis are defined. Based on the quantitative relations between the frequency coupling and the stability analysis of the grid-connected MMC system, a simple stability analysis method for the grid-connected MMC is proposed without losing accuracy. Specifically, in the cases where the coupling needs to be considered, the stability of MMC should be assessed by the GNC using the MIMO model, while for the other cases, the SISO model can be used.

The paper is organized as follows: The system model of the grid-connected MMC is given in Section 2. Section 3 derives the admittance matrix of MMC considering the frequency coupling, which is validated by experimental measurements. In Section 4, the coupling effect on the stability analysis of grid-connected MMC is analyzed. Section 5 draws the conclusions.

## 2. System Model

The topology of a three-phase MMC is indicated in Figure 1. Each phase of MMC includes an upper arm and a lower arm, and each arm consists of an $N$ submodule (SM) and arm inductor. Each SM constitutes an IGBT half-bridge and a capacitor $C_{SM}$.

The MMC is modeled in one phase, for example, which makes the subscript denoting the phase dropped when not needed. The phase current $i_g$ and circulating current $i_c$ are defined as:

$$\begin{cases} i_g(t) = i_u(t) - i_l(t) \\ i_c(t) = (i_u(t) + i_l(t))/2 \end{cases} \tag{1}$$

where $i_u(t)$ and $i_l(t)$ are the upper and lower arm currents.

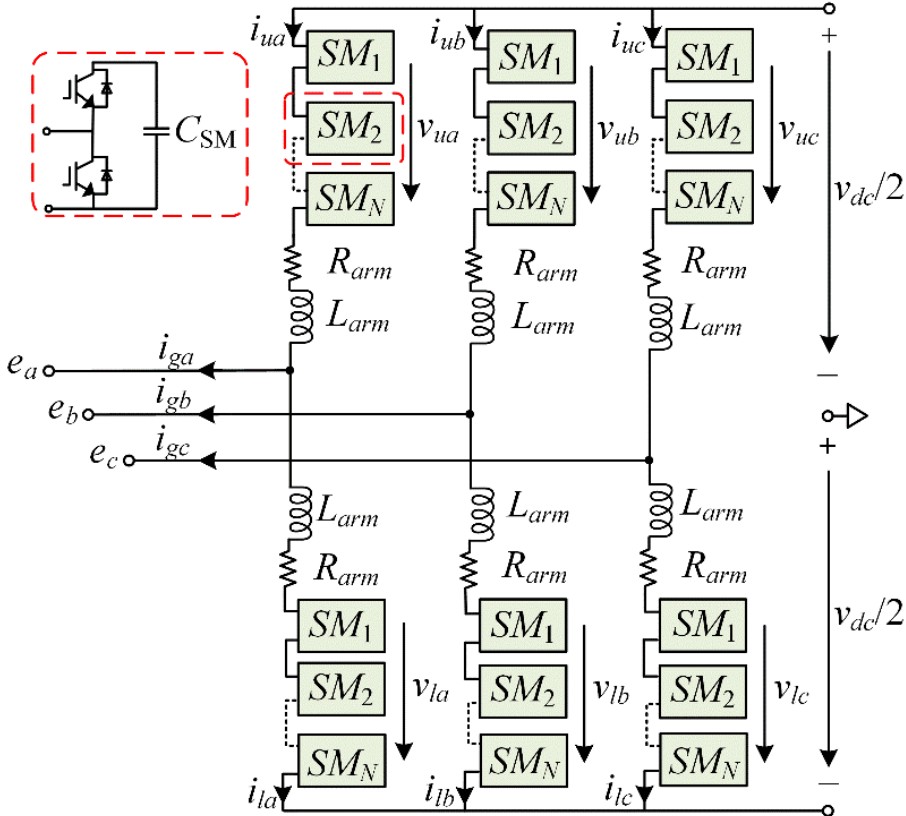

**Figure 1.** Topology of the modular multilevel converter.

Applying Kirchhoff's law, the current dynamics can be obtained as:

$$e(t) + L_{arm}\frac{di_u(t)}{dt} + R_{arm}i_u(t) + v_u(t) = v_{dc}/2 \tag{2}$$

$$e(t) - L_{arm}\frac{di_u(t)}{dt} - R_{arm}i_u(t) - v_l(t) = -v_{dc}/2 \tag{3}$$

where $v_u(t)$ and $v_l(t)$ are the upper and lower arm voltage, $e(t)$ is the point-of-common-coupling (PCC) voltage, and $v_{dc}$ is the DC-bus voltage.

This paper adopts the time-averaged model of MMC, where the switching operations of the SMs are neglected and the SM capacitor voltages are assumed to be balanced [31]. Thus, the individual SMs dynamics in the arm can be neglected. Based on the averaged model, the arm voltages $v_{u,l}(t)$ and the sum capacitor voltages $v_{Cu,l}^\Sigma(t)$ are obtained as:

$$v_{u,l}(t) = m_{u,l}(t)v_{Cu,l}^\Sigma(t) \tag{4}$$

$$C_{arm}\frac{dv_{Cu,l}^\Sigma(t)}{dt} = m_{u,l}(t)i_{u,l}(t) \tag{5}$$

where $C_{arm} = C_{SM}/N$ is the arm equivalent capacitance, and $m_{u,l}(t)$ are the modulation indices generated by the control scheme of MMC, expressed as:

$$\begin{cases} m_u(t) = (v_{dc}/2 - v_s(t) - v_c(t))/v_{dc} \\ m_l(t) = (v_{dc}/2 + v_s(t) - v_c(t))/v_{dc} \end{cases} \tag{6}$$

where $v_s(t)$ and $v_c(t)$ are the voltages from the phase current controller and circulating current controller, respectively.

Substituting Equations (1) and (4) into Equations (2) and (3) yields:

$$2e(t) + R_{arm}i_g(t) + L_{arm}\frac{di_g(t)}{dt} + m_u(t)v_{Cu}^{\Sigma}(t) - m_l(t)v_{Cl}^{\Sigma}(t) = 0 \tag{7}$$

$$2R_{arm}i_c(t) + 2L_{arm}\frac{di_c(t)}{dt} + m_u(t)v_{Cu}^{\Sigma}(t) + m_l(t)v_{Cl}^{\Sigma}(t) = v_{dc} \tag{8}$$

Equations (5), (7) and (8) represent the dynamic model of MMC.

Figure 2 shows the control scheme of the grid-connected MMC. The DC-bus voltage controller adopts the proportional-integral (PI) compensator to keep the DC-bus voltage as a constant and generate the reference current for the current control [32], which is shown in Figure 2a; $k_{pd}$ and $k_{id}$ are the proportional and integral coefficients of the dc voltage controller; the $q$-axis reference current is set to zero for unity power factor operation. The phase current controller adopts the PI compensator to properly track the reference currents, as illustrated in Figure 2b; $k_{pi}$ and $k_{ii}$ are the proportional and integral coefficients of the phase current controller. The circulating current suppression controller (CCSC) using the proportional resonant (PR) controller is shown in Figure 2c, which suppresses the 2nd harmonics in the circulating current [33]; $k_{pc}$ and $k_{rc}$ are the proportional and resonant coefficients of the CCSC. The phase-locked loop (PLL) synchronizes the MMC with the grid, as shown in Figure 2d; the proportional and integral coefficients of the PLL are $k_{ppll}$ and $k_{ipll}$. The voltages driving the grid current and circulating current ($u_s$ and $u_c$) are used for computing the modulation indices.

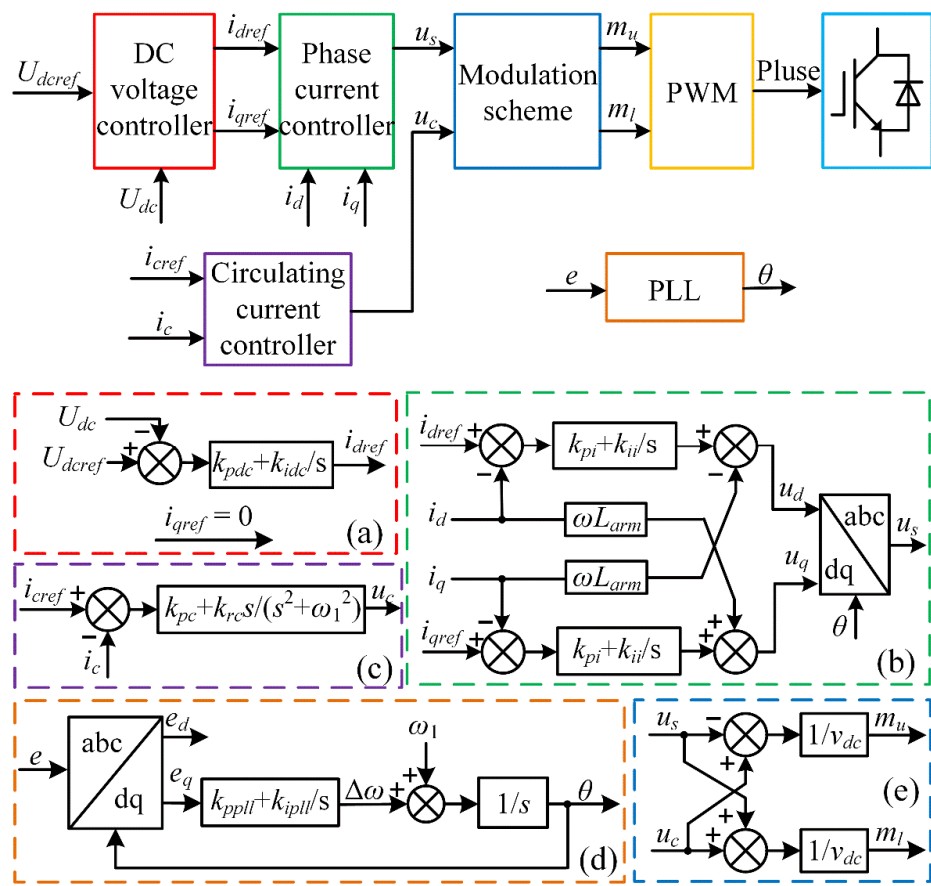

**Figure 2.** Control scheme of the grid-connected MMC. (**a**) DC-bus voltage control. (**b**) Phase current control. (**c**) Circulating current suppression controller. (**d**) PLL. (**e**) Modulation scheme.

### 3. Admittance Modeling of MMC Considering Frequency Coupling

*3.1. Small-Signal Model of MMC*

The AC-side admittance of MMC is obtained by injecting a perturbation voltage $e_p(t)$ at an arbitrary perturbation frequency $f_p$ in the PCC voltages at a fundamental frequency $f_1$, and calculating the corresponding current. The perturbation voltage is defined as:

$$e_p(t) = E_p \cos(2\pi f_p t) \tag{9}$$

where $E_p$ is the magnitude of the perturbation voltage.

Based on the harmonic linearization, the small-signal equations of the MMC in the time-domain are given as:

$$2e_p(t) + R_{arm}i_{gp}(t) + L_{arm}\frac{di_{gp}(t)}{dt} + m_{up}(t)v_{Cu0}^{\Sigma}(t)$$
$$+m_{u0}(t)v_{Cup}^{\Sigma}(t) - m_{lp}(t)v_{Cl0}^{\Sigma}(t) - m_{l0}(t)v_{Clp}^{\Sigma}(t) = 0 \tag{10}$$

$$2R_{arm}i_{cp}(t) + 2L_{arm}\frac{di_{cp}(t)}{dt} + m_{up}(t)v_{Cu0}^{\Sigma}(t)$$
$$+m_{u0}(t)v_{Cup}^{\Sigma}(t) + m_{lp}(t)v_{Cl0}^{\Sigma}(t) + m_{l0}(t)v_{Clp}^{\Sigma}(t) = 0 \tag{11}$$

$$C_{arm}\frac{dv_{Cup}^{\Sigma}(t)}{dt} = m_{up}(t)i_{u0}(t) + m_{u0}(t)i_{up}(t) \tag{12}$$

$$C_{arm}\frac{dv_{Clp}^{\Sigma}(t)}{dt} = m_{lp}(t)i_{l0}(t) + m_{l0}(t)i_{lp}(t) \tag{13}$$

where the subscript "$p$" indicates the small-signal variable and the subscript "0" represents the variable at the steady state.

*3.2. Frequency Coupling Analysis of MMC Admittance*

It is known that the positive-sequence perturbation voltage at $f_p$ will lead to the perturbation current at the same frequency in the power converter. As for MMC, the perturbation current at $f_p$ will generate perturbation capacitor fluctuation voltage $v_{Cp}^{\Sigma}$ at $f_1 \pm f_p$ due to the multiplication between the modulation index $m_1$ at $f_1$ and the perturbation current at $f_p$, as shown in Equations (12) and (13). Subsequently, the multiplication of $v_{Cp}^{\Sigma}$ at $f_1 \pm f_p$ with $m_1$ at $f_1$ leads to the perturbation arm voltages $v_p$ at $f_p$ and $f_p \pm 2f_1$. The perturbation arm voltages finally result in the perturbation arm currents at $f_p$ and $f_p \pm 2f_1$. Similarly, the perturbation modulation index at $f_p$ caused by the perturbation current will also generate the perturbation currents at $f_p$ and $f_p \pm 2f_1$. Considering the delay fundamental cycle caused by perturbation voltage $e_p$, the perturbation arm current at $f_p + 2f_1$ is a zero-sequence current, which will not appear in the three-phase three-line MMC system.

Based on the analysis above, it can be observed that the perturbation voltage at $f_p$ will lead to the perturbation currents at $f_p$ and $f_p - 2f_1$ for the MMC even under the symmetric control. The response of a linear and decoupled system contains only one component with the same frequency corresponding to the perturbation voltage. The additional component indicates the frequency coupling inherently exists in the MMC admittance, where the root cause is the capacitor voltage harmonics in MMC. The illustration of the frequency coupling is shown in Figure 3.

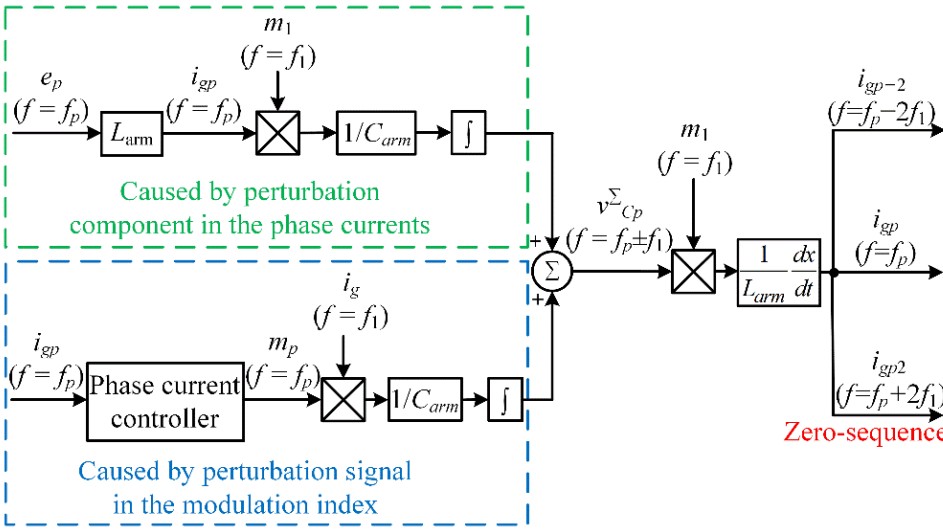

**Figure 3.** Illustration of the frequency coupling in MMC.

The sequence of the perturbation currents in MMC is shown in Figure 4. For the positive-sequence perturbation voltage at $f_p$, the corresponding current at $f_p$ is the positive-sequence component. The perturbation current at $f_p - 2f_1$ is yielded due to the capacitor fluctuation voltage in MMC. As can be seen from Figure 4, when the frequency $f_p$ is above $2f_1$, the perturbation current at $f_p - 2f_1$ is the negative-sequence component. While, if $f_p$ is below $2f_1$, the perturbation currents at $f_p - 2f_1$ is the negative-sequence component at negative frequency, which is equal to the positive-sequence current at positive frequency.

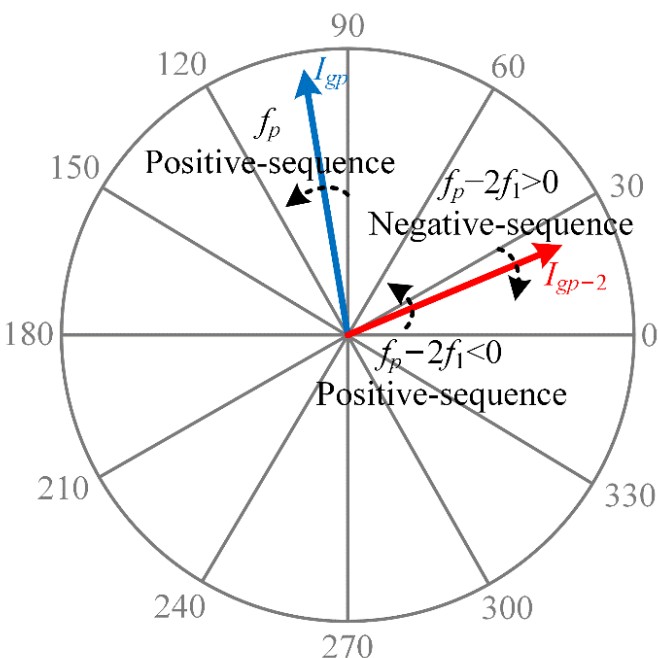

**Figure 4.** Sequence of the perturbation currents in MMC.

It should be noted that the perturbation voltage at $f_p$ will also interact with the state variables at $nf_1$ ($n \geq 3$) and generate the perturbation currents at $f_p \pm nf_1$. However, the amplitude of the perturbation currents is gradually decreased as the frequency increases. As shown in Figure 5, the amplitude of the harmonic currents at $f_p \pm nf_1$ ($n \geq 3$) is much smaller than that of the harmonic current at $f_p$, which can be ignored.

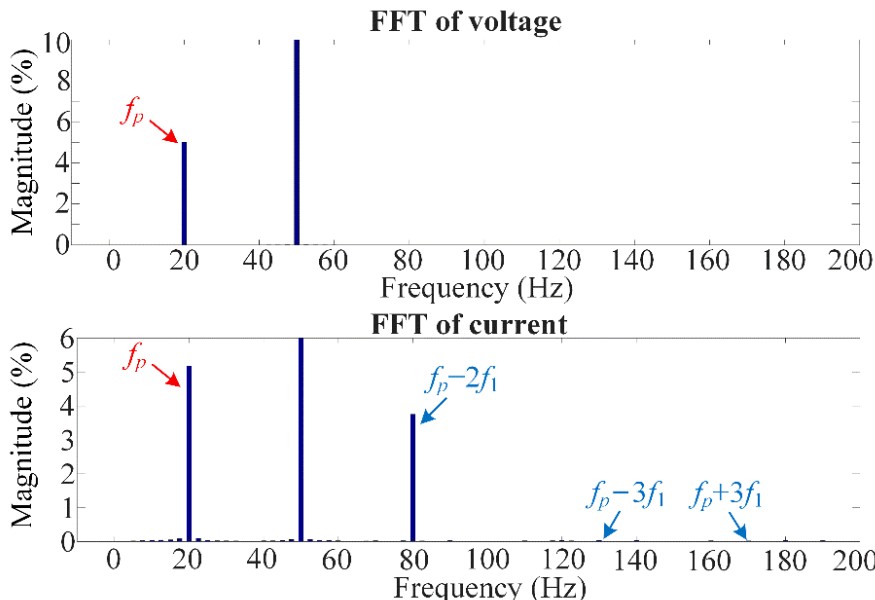

**Figure 5.** Frequency spectrum of the voltage and current with an injected perturbation voltage at 20 Hz.

### 3.3. Admittance Modeling

Due to the MMC internal dynamics, the multiple harmonics must be considered in the admittance modeling of MMC. There are several frequency-domain methods to derive the MMC admittance model (e.g., multiple-harmonic linearization [16,34], HSS modeling [14], harmonic transfer function [23], etc.). The essences of these methods are identical, which can simultaneously represent multiple frequency responses. Similar to these methods, the frequency-domain model of MMC which contains multiple perturbation components at $\omega_p, \omega_p \pm \omega_1, \omega_p \pm 2\omega_1, \ldots, \omega_p \pm n\omega_1$, is used in this paper.

The perturbation state variables in the time-domain can be expressed in the form of Fourier series as below:

$$x_p(t) = \sum_{n \in Z} X_{pn} e^{j(\omega_p + n\omega_1)t} \tag{14}$$

where $X_{pn}$ represents the Fourier coefficient corresponding to the frequency of $\omega_p \pm n\omega_1$.

Then, based on the Fourier series and harmonic balance theory [35], the time-domain model of the MMC in Equations (10)–(13) can be transformed to the frequency-domain model, which is expressed as:

$$s\boldsymbol{X}_p = (\boldsymbol{A}_p - \boldsymbol{N}_p)\boldsymbol{X}_p + \boldsymbol{B}_p\boldsymbol{U}_p \tag{15}$$

where:

$$\boldsymbol{X}_p = \left[X_{p-n}, \cdots, X_{p-1}, X_{p0}, X_{p1}, \cdots, X_{pn}\right]^T \tag{16}$$

$$X_{p\pm n} = \left[I_{gp\pm n}, I_{cp\pm n}, V^{\Sigma}_{Cup\pm n}, V^{\Sigma}_{Clp\pm n}\right]^T \tag{17}$$

$$\boldsymbol{U}_p = \left[U_{p-n}, \cdots, U_{p-1}, U_{p0}, U_{p1}, \cdots, U_{pn}\right]^T \tag{18}$$

$$U_{p\pm n} = \left[M_{up\pm n}, M_{pl\pm n}, E_{p\pm n}\right]^T \tag{19}$$

$\boldsymbol{A}_p$ and $\boldsymbol{B}_p$ are given in the Appendix A.

It should be noted that the perturbation modulation $M_{up\pm n}$ and $M_{lp\pm n}$ is related to the control dynamics. Thus, the control model should be obtained to get the admittance model. The detailed derivations of the control modeling are given in Appendix B.

Based on the control modeling in Appendix B, the perturbation modulation indices are expressed as:

$$\begin{cases} m_{up} = (H_g + Q)i_{gp} + H_c i_{cp} + (P + T)e_p \\ m_{lp} = -(H_g + Q)i_{gp} + H_c i_{cp} - (P + T)e_p \end{cases} \tag{20}$$

Substituting the control model under positive-sequence perturbation voltage into the small-signal model in the frequency-domain, the perturbation state variables can be calculated by:

$$X_p = (A_p - N_p)^{-1} B_p U_p \tag{21}$$

Extracting the perturbation currents at $f_p$ and $f_p - 2f_1$ from the perturbation state variables, the positive-sequence admittance and the coupling term of MMC can be calculated as:

$$Y_p(s) = -I_{gp0}/E_p, C_{p-2}(s - 2j\omega_1) = -I_{gp-2}/E_p \tag{22}$$

Similarly, the negative-sequence admittance and the coupling term of MMC can be calculated by substituting the control model under negative-sequence voltage perturbation into the small-signal model, which is obtained as:

$$Y_n(s - 2j\omega_1) = -I_{gp-2}/E_{p-2}, C_{n+2}(s) = -I_{gp0}/E_{p-2} \tag{23}$$

where $E_{p-2}$ is the complex phasors of the negative-sequence perturbation voltage at $f_p - 2f_1$, and $I_{gp0}$ is the complex phasors of the perturbation current at $f_p$.

Considering the frequency coupling, the admittance matrix of MMC can be defined as:

$$Y_{\text{MMC}}(s) = \begin{bmatrix} Y_p(s) & C_{n+2}(s) \\ C_{p-2}(s - 2j\omega_1) & Y_n(s - 2j\omega_1) \end{bmatrix} \tag{24}$$

where the diagonal elements represent the sequence admittance, whereas the off-diagonal elements represent the coupling terms.

### 3.4. Experimental Verification

To validate the MMC admittance model, the admittance measurements are carried out on a down-scaled MMC prototype. The main parameters of MMC are listed in Table 1. The configuration and photograph of the experimental setup for measuring the MMC admittance are shown in Figures 6 and 7. The grid simulator, acting as the grid voltage source and the perturbation voltage injection source, injects the perturbation voltages ranging from 1 to 1000 Hz in one-phase (phase b) voltage. Moreover, in order to acquire the components of the admittance matrix in Equation (24), two linearly independent perturbation voltages at $f_p$ and $f_p - 2f_1$ are required to gain enough information [36]. The perturbation voltage $e_{p1} = E_p \cos(2\pi f_p - 2\pi/3)$ is firstly injected while the perturbation voltage at $f_p - 2f_1$ is equal to zero. Similarly, a second perturbation sequence is implemented by injecting the perturbation voltage $e_{p2} = E_p \cos[2\pi(f_p - 2f_1) + 2\pi/3]$ with the perturbation voltage at $f_p$ being zero. Then, the perturbation voltage and resulting current are extracted from the measured voltage and current by FFT. When the perturbation voltage at $f_p$ is superposed to the voltage at $f_1$, the response currents at $f_p$ and $f_p - 2f_1$ are only caused by $e_{p1}(f_p)$ and the admittance $Y_p$ and $C_{p-2}$ are identified as:

$$Y_p(s) = -\frac{i_{gb}(f_p)}{e_b(f_p)}\bigg|_{e_{p2}(f_p - 2f_1) = 0}, C_{p-2}(s) = -\frac{i_{gb}(f_p - 2f_1)}{e_b(f_p)}\bigg|_{e_{p2}(f_p - 2f_1) = 0} \tag{25}$$

**Table 1.** Parameters of MMC in the downscaled prototype.

| Symbol | Description | Value |
|--------|-------------|-------|
| $V_s$ | Line-to-line grid voltage (RMS) | 380 V |
| $L_{arm}$ | Arm inductance | 5 mH |
| $v_{dc}$ | DC-bus voltage of MMC | 800 V |
| $N$ | Modules per arm | 2 |
| $C_{SM}$ | Submodule capacitance | 1 mF |
| $f_1$ | Fundamental frequency | 50 Hz |
| $k_{pi}/k_{ii}$ | Phase current controller | 0.5/5 |

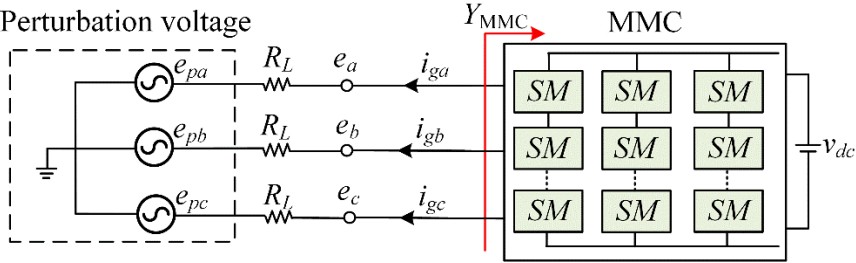

**Figure 6.** Configuration of the experimental setup for measuring MMC admittance.

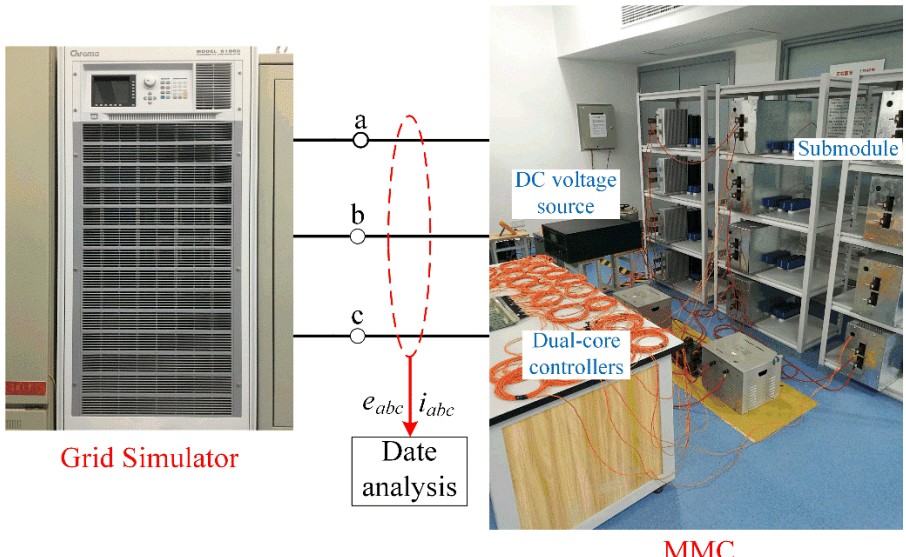

**Figure 7.** Photograph of the experimental setup.

Similarly, when the perturbation voltage at $f_p - 2f_1$ is injected, the admittance $Y_n$, $C_{n+2}$ can also be identified as:

$$Y_n = -\left.\frac{i_{gb}(f_p - 2f_1)}{e_b(f_p - 2f_1)}\right|_{e_{p1}(f_p)=0}, C_{n+2} = -\left.\frac{i_{gb}(f_p)}{e_b(f_p - 2f_1)}\right|_{e_{p1}(f_p)=0} \tag{26}$$

Note that if $f_p$ is below $2f_1$, the perturbation components at $f_p - 2f_1$ is the component at positive frequency with the opposite phase angle. The frequency and the phase angle of the perturbation components should be revised; the detailed process of the revision has been described in [37].

Figure 8 shows the comparison between the analytical admittance model and the experimental measurements of the MMC. The sequence analytical admittance models of MMC are denoted by the blue line, and the red line denotes the coupling terms in the MMC admittance matrix model. The measured values are denoted by the dots. The comparison in Figure 8 shows a good agreement between the analytical admittance models and the experimental measurements. This agreement validates the accuracy of the MMC admittance model. It should be noted that the magnitude of coupling terms (blue line) is close to that of sequence admittance (red line) at low frequency. Yet, the magnitudes of coupling terms are much smaller than that of sequence admittance above 100 Hz, which indicates the phenomenon of frequency coupling is not obvious at high frequency.

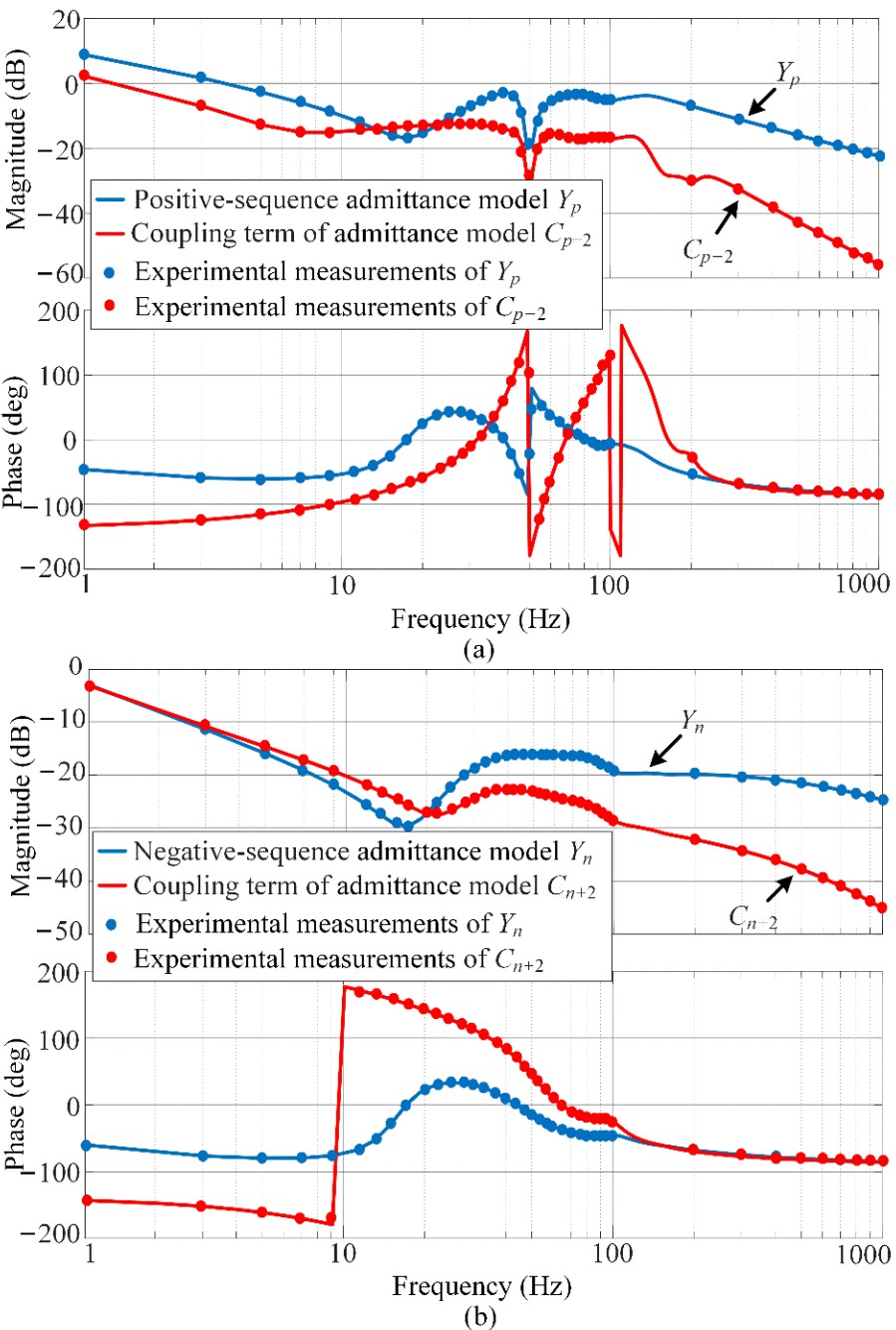

**Figure 8.** The theoretical and experimental validations of the admittance responses with the phase current controller only (**a**) $Y_p$ and $C_{p-2}$. (**b**) $Y_n$ and $C_{n+2}$. Notes that blue and red lines are theoretical results, the dots are from experimental measurements.

## 4. Effect of Frequency Coupling on Stability Analysis of Grid-Connected MMC

Figure 9 shows the small-signal description of the grid-connected MMC system. Assuming the grid is stable without the MMC, and the MMC is stable when connected to an ideal grid, the stability of the grid-connected MMC system can be assessed by the impedance ratio $Z_g(s)Y_{\mathrm{MMC}}(s)$, which should satisfy the Nyquist stability criterion.

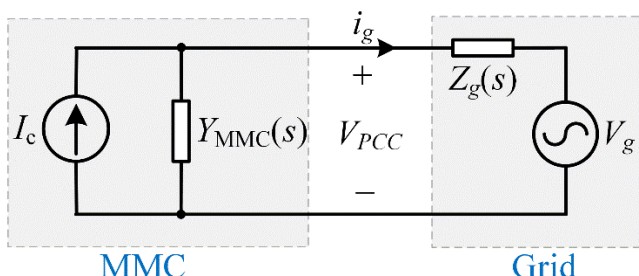

**Figure 9.** Small-signal representation of the grid-connected MMC system.

Due to the frequency coupling in the MMC admittance, the generalized Nyquist criterion (GNC) should be applied to evaluate the system stability [38]. The grid weakness is distinguished by the short circuit ratio (SCR) as it is an inverse of the grid impedance. In general, SCR $\geq 3$ denotes a strong system; $2 \leq$ SCR $< 3$ denotes a weak system; and $1 \leq$ SCR $< 2$ indicates a very weak system [39,40]. Then, the stability can be examined by checking the Nyquist plot of $Z_g(s)Y_{\mathrm{MMC}}(s)$, i.e., $\lambda_1$, $\lambda_2$, which are obtained as:

$$\begin{pmatrix} \lambda_1 \\ \lambda_2 \end{pmatrix} = \det\big(sI - \mathbf{Z}_g(s)\mathbf{Y}_{MMC}(s)\big) \tag{27}$$

where:

$$\mathbf{Z}_g(s) = \begin{bmatrix} Z_g(s) & \\ & Z_g(s - 2j\omega_1) \end{bmatrix} \tag{28}$$

Based on Equation (27), it can be concluded that if the coupling terms are ignored in the stability analysis, a deviation will be generated in the stability assessment, which is calculated as:

$$\Delta\lambda = \sqrt{Z_g(s)Z_g(s - 2j\omega_1)C_p(s)C_{p-2}(s - 2j\omega_1)} \tag{29}$$

According to Equation (29), the deviation $\Delta\lambda$ is proportional to the coupling terms. As can be seen from Figure 8, the magnitude of coupling terms is close to that of sequence admittance below 100 Hz, which leads to the relatively high deviation in the stability assessment. Thus, ignoring the frequency coupling has an influence on the stability assessment of grid-connected MMC under low frequency.

The effect of frequency coupling on the stability analysis of grid-connected MMCs can be evaluated following the steps shown in Figure 10. As seen from Figure 10, the SCR is initialized with 0.1. When the coupling is considered, the critical stability point $SCR_1$ increases with a fixed step during every iteration. The critical stability point $SCR_2$ with frequency coupling ignored is obtained in a similar way. Thus, if SCR belongs to [$SCR_1$, $SCR_2$], neglecting the frequency coupling leads to the incorrect stability analysis. If SCR does not belong to [$SCR_1$, $SCR_2$], the same stability analysis can be obtained, even though the frequency coupling is not considered. Based on the above analysis, the effect of frequency coupling on the stability analysis under different grid conditions can be evaluated.

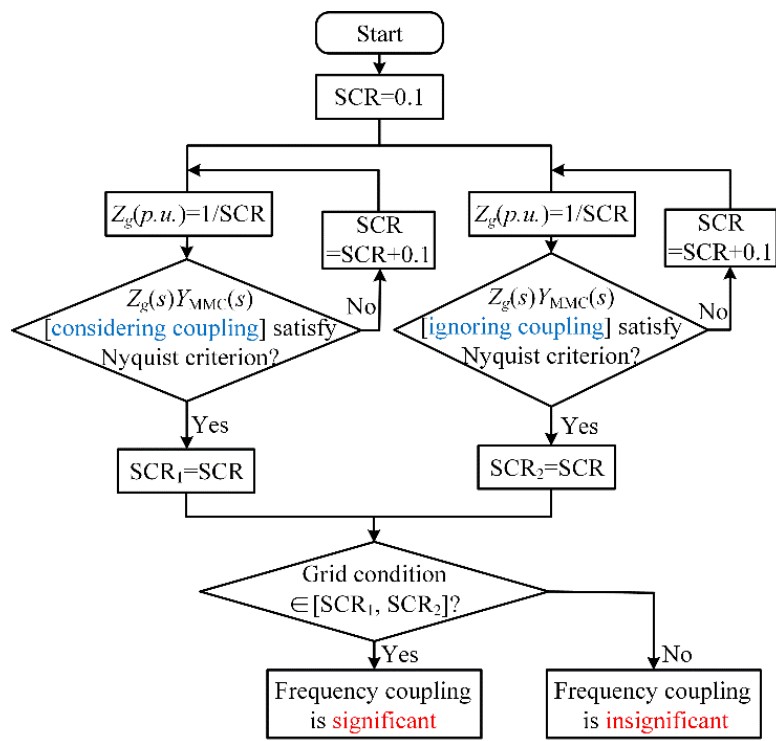

**Figure 10.** Flow chart for significance evaluation of frequency coupling.

An example MMC in China is analyzed in this paper to account for the coupling effect on the system stability analysis. The parameters of MMC are shown in Table 2. Due to the high power and a large number of submodules in the MMC-HVDC system, the simulation in MATLAB/Simulink is considered to be an acceptable method to verify the system-level stability analysis. Starting from the MMC operated with the phase current controller, the different controller elements are progressively added to analyze the frequency coupling effect on the stability analysis of the MMCs under different control schemes.

**Table 2.** Parameters of the practical MMC project.

| Symbol | Description | Value |
|--------|-------------|-------|
| $V_s$ | Line-to-line grid voltage (RMS) | 500 kV |
| $L_{arm}$ | Arm inductance | 50 mH |
| $v_{dc}$ | DC-bus voltage of MMC | $\pm$535 kV |
| $N$ | Modules per arm | 250 |
| $C_{SM}$ | Submodule capacitance | 15 mF |
| $f_1$ | Fundamental frequency | 50 Hz |
| $k_{pdc}/k_{idc}$ | DC voltage controller | 0.01/0.3 |
| $k_{pi}/k_{ii}$ | Phase current controller | 0.1/1 |
| $k_{pc}/k_{rc}$ | Circulating current suppression controller | 1/10 |
| $k_{ppll}/k_{ipll}$ | PLL | $1.2 \times 10^{-4}/7 \times 10^{-3}$ |

### 4.1. Case of MMC with Phase Current Controller

The Bode diagrams of the sequence admittance and coupling term of the MMC with the phase current controller are shown in Figure 11, where only the magnitude of $Y_p$ and $C_{p-2}$ are shown. It can be seen that the magnitude of the coupling term is close to that of the sequence admittance under 20 Hz, which indicates that the frequency coupling of MMC has a significant influence on the stability analysis of the grid-connected MMC system under 20 Hz.

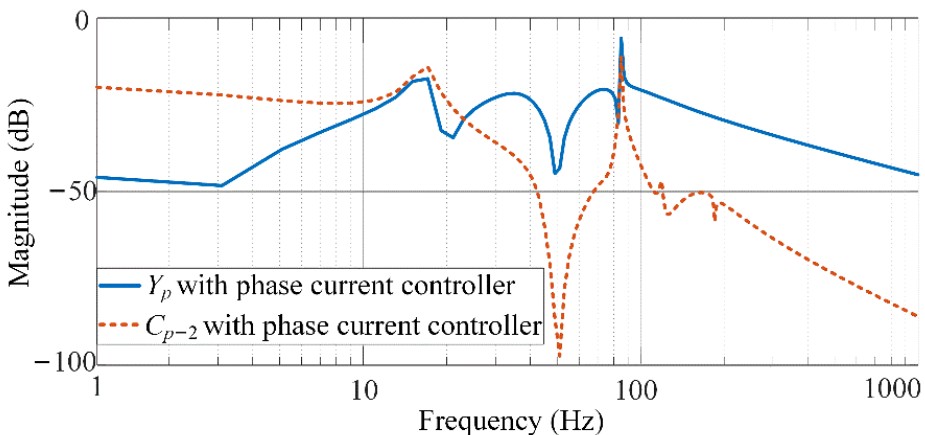

**Figure 11.** Bode diagrams of MMC admittance with the phase current controller.

With the iteration shown in Figure 10, the condition for considering the coupling is obtained as 1.9 < SCR < 2.6. Figure 12 shows the Nyquist plot of $\boldsymbol{Z_g}(s)\boldsymbol{Y}_{\text{MMC}}(s)$ considering or ignoring the frequency coupling, where only the dominant eigenvalue ($\lambda_1$) is shown for simplicity. It can be seen when the coupling is considered, the Nyquist plot encircles the point $(-1, j0)$ under SCR < 2.6, meaning that the system is unstable if SCR < 2.6. However, the Nyquist plot ignoring the coupling does not encircle the point $(-1, j0)$ until SCR is less than 1.9, indicating the system is predicted to be unstable when SCR < 1.9. Therefore, ignoring the coupling will lead to a wrong stability assessment if 1.9 < SCR < 2.6.

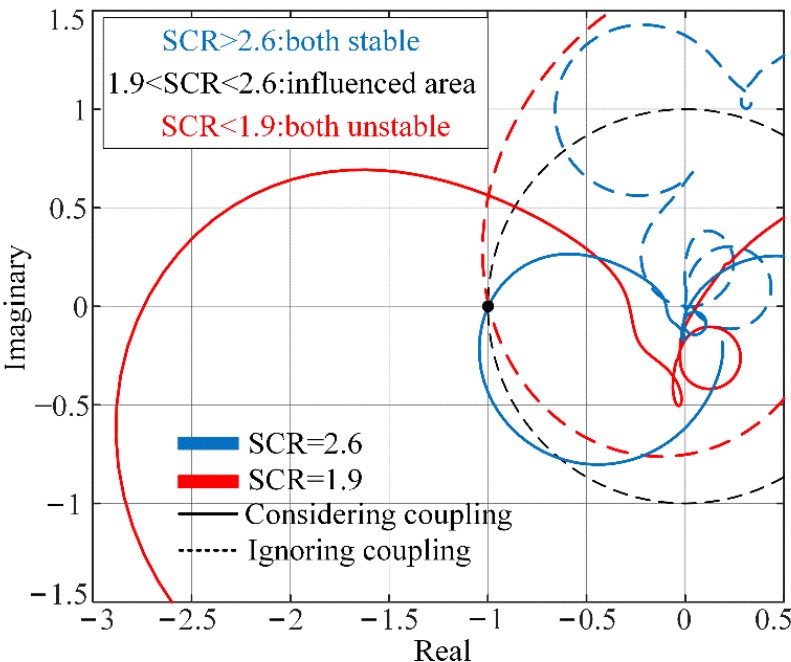

**Figure 12.** Nyquist plot of the impedance ratio $\boldsymbol{Z_g}(s)\boldsymbol{Y}_{\text{MMC}}(s)$ with the phase current controller.

Figure 13 shows the time-domain simulation results of the MMC under a step change of the grid condition, where only the phase current controller is adopted. The parameters in the simulation are the same as those in Table 2. It can be seen that the system is stable when SCR = 2.7. However, the grid-connected MMC loses stability as SCR = 2.5, which is consistent with the theoretical analysis in Figure 12. The dominant oscillation frequency is around 18 and 82 Hz. It is the coupling effect that causes both the sub- and super-synchronous frequency in the oscillation.

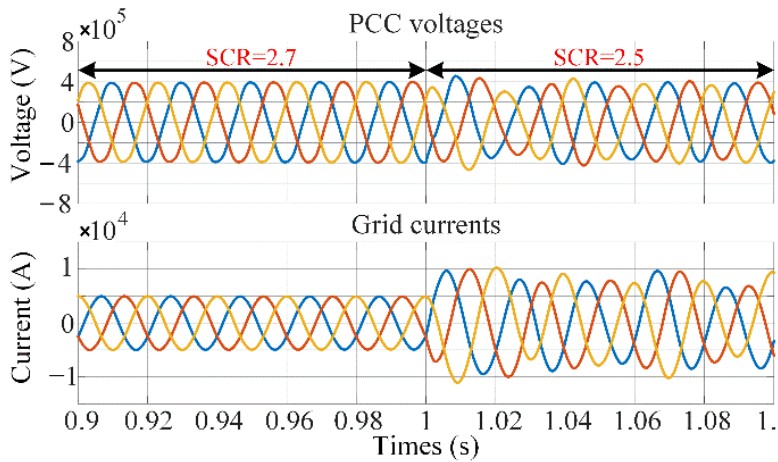

**Figure 13.** Simulation results of the grid-connected MMC with the phase current controller.

### 4.2. Case of MMC with CCSC

For this case, only the phase current controller and CCSC are applied in the control scheme. The Bode diagrams of the sequence admittances and coupling terms of MMC with and without the CCSC are shown in Figure 14 (only magnitude plots are shown). It can be seen that the CCSC can suppress the low-frequency resonance characteristics of both the sequence admittance and the coupling terms of MMC. Yet, the frequency coupling still exists in the MMC admittance, which indicates that the CCSC cannot eliminate the frequency coupling of the MMC system.

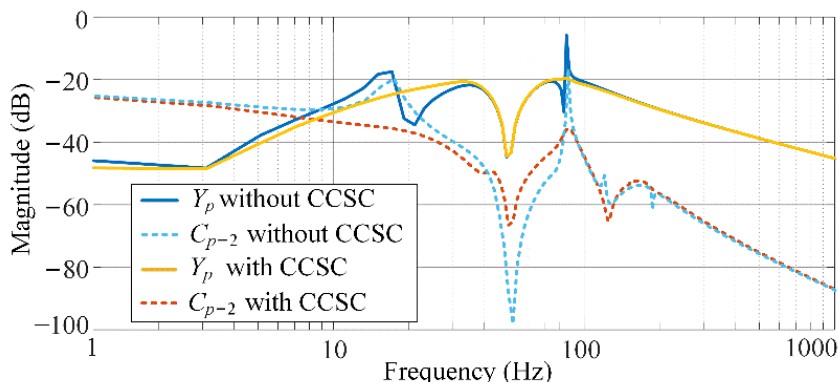

**Figure 14.** Bode diagrams of MMC admittances with and without CCSC.

Figure 15 shows the Nyquist plot of $Z_g(s)Y_{\mathrm{MMC}}(s)$ with CCSC, where the SCR = 1.5. As shown in Figure 15, both the stability assessments considering or ignoring the frequency coupling indicate that the system is stable even under the ultra-weak grid. The phenomenon can be explained as follows: As for the MMC with CCSC, the capacitive behavior of MMC impedance caused by its internal dynamics has been suppressed, which makes the stability margin of grid-connected MMCs sufficient even under a very weak grid. For the system with a sufficient stability margin, ignoring the coupling will not lead to a wrong stability assessment. Therefore, for the MMC only with the phase current controller and CCSC, ignoring the frequency coupling will not affect the stability analysis of the grid-connected MMC.

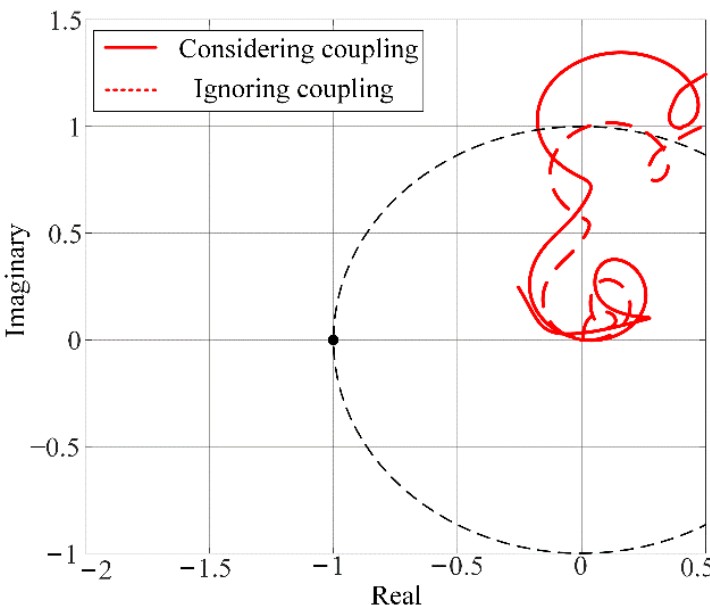

**Figure 15.** Nyquist plot of the impedance ratio $\mathbf{Z_g}(s)\mathbf{Y_{MMC}}(s)$ with CCSC, SCR = 1.5.

### 4.3. Case of MMC Considering PLL Dynamics

For this case, the phase current controller, CCSC, and PLL are adopted in MMC. Figure 16 shows the PLL effect on the sequence admittances and coupling terms of MMC. It can be observed that the PLL mostly affects the sequence admittances and coupling terms around the fundamental frequency. Furthermore, it is noted that except for the fundamental frequency, the coupling term at another frequency is barely affected by PLL.

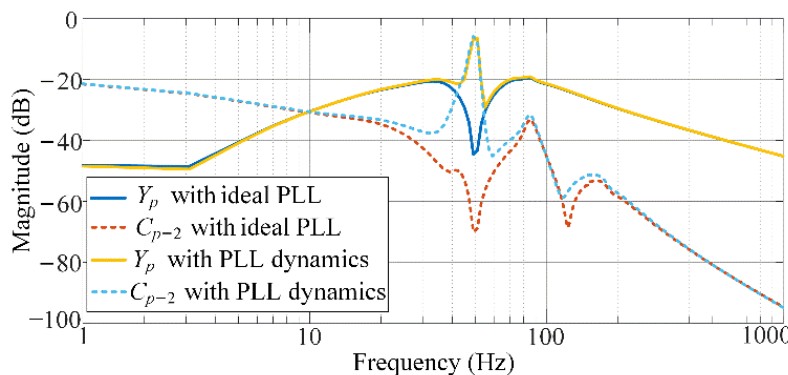

**Figure 16.** Bode diagrams of the MMC admittances with ideal PLL and PLL dynamics.

Substituting the MMC admittance model considering PLL dynamics into the iterative procedure in Figure 10, the conditions for considering the frequency coupling can be obtained. The results show that when SCR < 2.2, the stability assessments are unstable regardless of whether the coupling is ignored or considered. In addition, when SCR > 2.8, the system is always stable regardless of whether the coupling is ignored or considered. However, if the frequency coupling is ignored at 2.2 < SCR < 2.8, an inaccurate stability analysis will be obtained. Figure 17 shows the characteristic loci of $\mathbf{Z_g}(s)\mathbf{Y_{MMC}}(s)$ with PLL dynamics. The grid condition of 2.2 < SCR < 2.8 is the condition that ignoring the coupling will result in a wrong stability estimation.

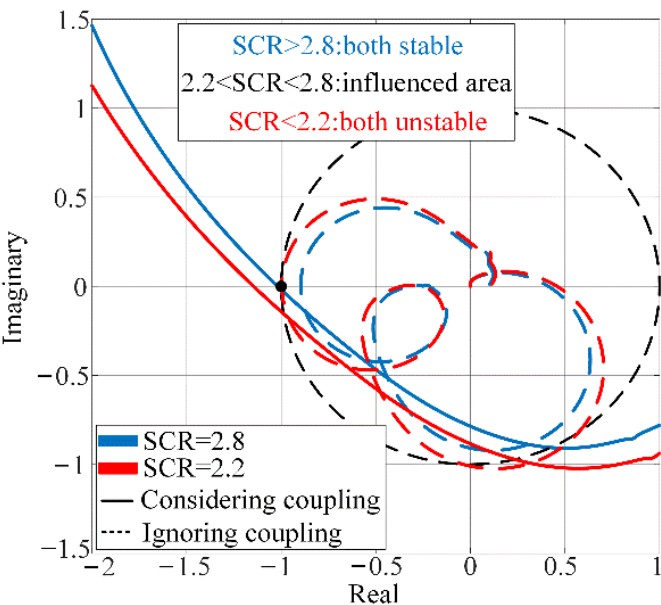

**Figure 17.** Nyquist plot of the impedance ratios $Z_g(s)Y_{MMC}(s)$ with PLL dynamics.

Figure 18 shows the time-domain simulation results of the MMC under a step change of the grid condition, where the PLL bandwidth is set as 40 Hz. It can be seen the system loses stability when SCR = 2.7.

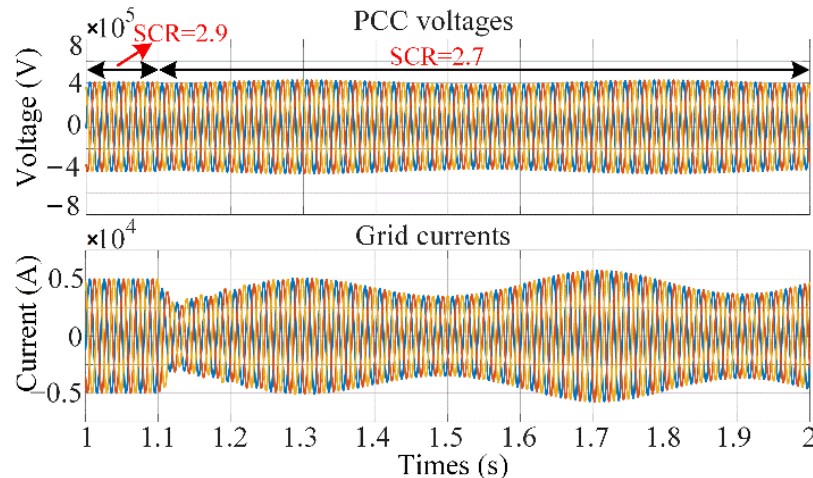

**Figure 18.** Simulation results of the grid-connected MMC with PLL dynamics.

For this case, the DC-voltage control is added in the case of MMC considering PLL dynamics. Figure 19 shows the effect of DC-bus voltage control on the MMC admittance. It can be seen that the DC-bus voltage control mainly affects the admittance responses of MMC below 10 Hz. Considering that adding DC voltage control hardly changes the MMC admittance characteristics, the stability analysis considering voltage control is almost the same as the stability analysis considering PLL dynamics. Therefore, the condition for considering the coupling is the same as that of MMC considering the PLL dynamics, which is not analyzed in detail.

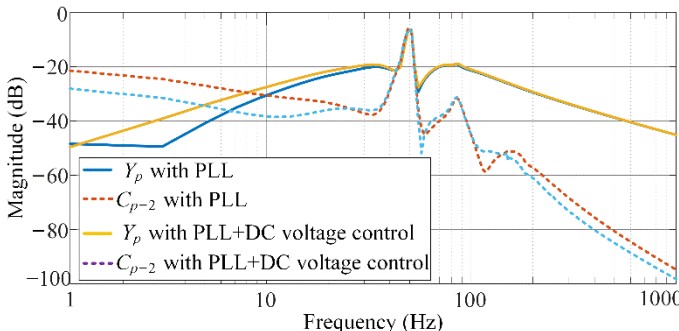

**Figure 19.** Bode diagrams of the MMC admittances with and without the DC-bus voltage controller.

*4.4. Discussion*

Based on the above analysis, it can be observed that ignoring the frequency coupling of MMC admittance leads to the inaccurate stability conclusion under certain grid conditions. However, it is not always the case that ignoring coupling fails in the stability assessment. The conditions for considering the coupling effect are quantitatively defined in Section 4 (1–4). Table 3 summarizes the conditions for different cases of MMC.

**Table 3.** Significance evaluation of frequency coupling.

| Significance of Coupling | Significant | Insignificant |
| --- | --- | --- |
| Phase current controller | 1.9 < SCR < 2.6 | SCR < 1.9, SCR > 2.6 |
| Phase current controller+ CCSC | none | none |
| Phase current controller+ CCSC +PLL | 2.2 < SCR < 2.8 | SCR < 2.2, SCR > 2.8 |
| Phase current controller+ CCSC +PLL + DC-bus voltage controller | 2.2 < SCR < 2.8 | SCR < 2.2, SCR > 2.8 |

Previous studies have analyzed the stability of an MMC system using the SISO impedance model [14,15]. Yet, the SISO model cannot include the impact of frequency coupling on the stability, which may lead to inaccurate results in some cases. Afterward, the MIMO modeling method, which considers the frequency coupling effect, is proposed. Compared with the SISO model, the MIMO model can guarantee the accuracy of stability analysis results under any circumstances. However, the stability analysis method applying MIMO (e.g., GNC) is more complicated than the stability analysis method applying on the SISO model (e.g., Bode plot). Meanwhile, it is not always the case that the SISO model fails in the stability assessment of the grid-connected MMC. Thus, different models can be applied to the stability of MMCs under different grid conditions, as shown in Figure 20. According to the significance evaluation of frequency coupling and the practical grid condition, it can be determined whether the MIMO model of MMC impedance needs to be applied in the stability assessment. In the cases where the coupling needs to be considered, the stability of MMC should be assessed by the GNC using the MIMO model, while for the other cases, the SISO model can be used. Compared to the stability analysis method that always applies the MIMO model, the proposed method is simpler without losing accuracy.

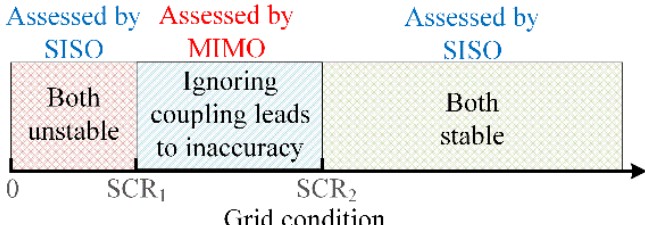

**Figure 20.** Method of the stability assessment of the grid-connected MMC system.

## 5. Conclusions

Ignoring the frequency coupling will lead to inaccuracy in stability analysis for the grid-connected MMC under a certain short circuit ratio of the grid. However, it is not always the case that ignoring coupling fails in the stability assessment. In order to clarify the case that the coupling must be considered, the effect of frequency coupling of MMC impedance on the stability analysis of grid-connected MMC was quantitatively analyzed in this paper. On this basis, a stability analysis method was proposed, where GNC is applied when the frequency coupling has a significant effect and the SISO analysis tool (e.g., Bode plot) is used if the coupling is insignificant. Compared to the stability assessment always applying the MIMO model, the proposed method is more simplified without losing accuracy. The grid impedance is different for the grids under different active power. Thus, the grid conditions that the coupling effect needs to be considered in the stability analysis are different for the grids under different active power. The proposed method can be only applied for a grid with a fixed power. The method suitable for the grids under different active power will be studied in the future.

**Author Contributions:** Conceptualization, Y.W., Q.X.; methodology, Y.W., Q.X.; software, Y.W.; validation, Y.W., Q.X.; formal analysis, Y.W., Q.X.; investigation, Y.W., Q.X.; resources, Q.X.; data curation, Y.W.; writing—original draft preparation, Y.W.; writing—review and editing, Y.W.; visualization, Y.W.; supervision, J.M.G.; project administration, Q.X.; funding acquisition, Q.X. All authors have read and agreed to the published version of the manuscript.

**Funding:** This research was funded by the National Natural Science Foundation of China (51807056).

**Institutional Review Board Statement:** Not applicable.

**Informed Consent Statement:** Not applicable.

**Data Availability Statement:** Not applicable.

**Conflicts of Interest:** The authors declare no conflict of interest.

## Appendix A

The expression of $A_p$ and $B_p$:

$$A_p = \Gamma(A_p) = \begin{bmatrix} A_{p0} & A_{p-1} & \cdots & A_{p-n} & & & \\ A_{p1} & \ddots & \ddots & \ddots & \ddots & & \\ \vdots & \ddots & A_{p0} & A_{p-1} & \ddots & \ddots & \\ A_{pn} & \ddots & A_{p1} & A_{p0} & A_{p-1} & \ddots & A_{p-n} \\ & \ddots & \ddots & A_{p1} & A_{p0} & \ddots & \vdots \\ & & \ddots & \ddots & \ddots & \ddots & A_{p-1} \\ & & & A_{pn} & \cdots & A_{p1} & A_{p0} \end{bmatrix} \tag{A1}$$

$$A_p = \begin{bmatrix} -\frac{R_{arm}}{L_{arm}} & 0 & -\frac{m_{u0}(t)}{L_{arm}} & \frac{m_{l0}(t)}{L_{arm}} \\ 0 & -\frac{R_{arm}}{L_{arm}} & -\frac{m_{u0}(t)}{2L_{arm}} & -\frac{m_{l0}(t)}{2L_{arm}} \\ \frac{m_{u0}(t)}{2C_{arm}} & \frac{m_{u0}(t)}{C_{arm}} & 0 & 0 \\ -\frac{m_{l0}(t)}{2C_{arm}} & \frac{m_{l0}(t)}{C_{arm}} & 0 & 0 \end{bmatrix} \tag{A2}$$

$$B_p = \Gamma(B_p) \tag{A3}$$

$$B_p = \begin{bmatrix} -\dfrac{v_{Cu0}^{\Sigma}(t)}{L_{arm}} & \dfrac{v_{Cl0}^{\Sigma}(t)}{L_{arm}} & -\dfrac{2}{L_{arm}} \\ -\dfrac{v_{Cu0}^{\Sigma}(t)}{2L_{arm}} & -\dfrac{v_{Cl0}^{\Sigma}(t)}{2L_{arm}} & 0 \\ -\dfrac{0.5i_{g0}(t)+i_{c0}(t)}{C_{arm}} & 0 & 0 \\ 0 & \dfrac{-0.5i_{g0}(t)+i_{c0}(t)}{C_{arm}} & 0 \end{bmatrix} \tag{A4}$$

$$N_p = diag\big[-j(\omega_p - n\omega_1)\boldsymbol{I}, \cdots, \boldsymbol{O}, \cdots, j(\omega_p + n\omega_1)\boldsymbol{I}\big] \tag{A5}$$

where $\boldsymbol{I}$ is the identity matrix and $\boldsymbol{O}$ denotes the zero matrix. $\boldsymbol{A}$ and $\boldsymbol{B}$ are the Toeplitz ($\Gamma$) matrixes using as an alternative of the convolution. The steady-state values in (A1)~(A4) can be obtained by solving MMC nonlinear equations, or the simulation of the converter circuit.

**Appendix B**

The basic MMC control functions (e.g., phase current controller, circulating current suppression controller, and PLL) have been modeled in [16], which are expressed as follows.

(1) Phase current controller:

The phase current controller model is:

$$\boldsymbol{m}_{up} = \boldsymbol{H}_g\boldsymbol{i}_{gp}, \boldsymbol{m}_{lp} = -\boldsymbol{H}_g\boldsymbol{i}_{gp}; \boldsymbol{H}_g = diag\Big[\{q_k\}|_{k=-n,\cdots,0,\cdots,n}\Big]$$
$$q_k = \frac{1+(-1)^k}{2}|mod(k+1,3)| \cdot \big[-mod(k+1,3)jK_d + H_i\big(j(\omega_p + k\omega_1) - mod(k+1,3)j\omega_1\big)\big] \tag{A6}$$

where:

$$\boldsymbol{m}_{up} = \big[M_{up-n}, \cdots, M_{up-1}, M_{up0}, M_{up1}, \cdots, M_{upn}\big]^T \tag{A7}$$

$\boldsymbol{m}_{lp}$ follows a similar expression. $H_i$ is the transfer function of the current controller in $dq$-frame, the $jK_d$ term indicates the effect of the decoupling gain. The term $mod(k+1,3)$ is the modulo-3 function, which gives the sequence of the small-signal perturbation current.

Under a negative-sequence perturbation voltage, $\boldsymbol{H}_g$ can be expressed as:

$$\boldsymbol{H}_g = diag\big[\{q_k\}|k=-n,\cdots,0,\cdots,n\big]$$
$$q_k = \frac{1+(-1)^k}{2}|mod(k-1,3)| \cdot \big[-mod(k-1,3)jK_d + H_i\big(j(\omega_p + k\omega_1) - mod(k-1,3)j\omega_1\big)\big] \tag{A8}$$

(2) Circulating current suppression controller

The model of CCSC is expressed as:

$$\boldsymbol{m}_{up} = \boldsymbol{H}_c\boldsymbol{i}_{cp}, \boldsymbol{m}_{lp} = \boldsymbol{H}_c\boldsymbol{i}_{cp}; \boldsymbol{H}_c = diag\Big[\{q_k\}|_{k=-n,\cdots,0,\cdots,n}\Big]$$
$$q_k = \frac{1-(-1)^k}{2}H_c\big(j(\omega_p + k\omega_1)\big) \tag{A9}$$

where $H_c$ is the transfer function of PR compensator.

(3) PLL

The PLL model is expressed as:

$$\boldsymbol{m}_{up} = \boldsymbol{P}\boldsymbol{e}_p, \boldsymbol{m}_{lp} = -\boldsymbol{P}\boldsymbol{e}_p \tag{A10}$$

where $\boldsymbol{P}$ is a $(2n+1) \times (2n+1)$ matrix, $(n+1, n+1)$th element is:

$$G_\theta\big(j(\omega_p - \omega_1)\big) \cdot \big(\big[H_i\big(j(\omega_p - \omega_1)\big) + jK_d\big]I_g - M_1\big) \tag{A11}$$

and the $(n+1, n-1)$th element is:

$$-G_\theta\big(j(\omega_p - \omega_1)\big) \cdot \big(\big[H_i\big(j(\omega_p - \omega_1)\big) - jK_d\big]I_g{}^* - M_1{}^*\big) \tag{A12}$$

where $I_g$ and $M_1$ represent the Fourier series of the phase current and the modulation index at steady state. $I_g^*$ and $M_1^*$ are the complex conjugates of $I_g$ and $M_1$. $G_\theta$ is the transfer function expressing PLL dynamics, which is given as:

$$G_\theta\big(j(\omega_p - \omega_1)\big) = T_\theta\big(j(\omega_p - \omega_1)\big) / \big(1 + E_1 T_\theta\big(j(\omega_p - \omega_1)\big)\big),$$
$$T_\theta\big(j(\omega_p - \omega_1)\big) = H_{PI}\big(j(\omega_p - \omega_1)\big) / j(\omega_p - \omega_1) \tag{A13}$$

where $E_1$ is the steady-state grid voltage and $H_{PI}$ is the transfer function of the PI compensator in PLL.

For a negative-sequence perturbation, **P** is a $(2n + 1) \times (2n + 1)$ matrix with $(n + 1, n + 1)$th element and $(n + 1, n + 3)$th element. The elements are obtained by replacing $f_p$ in (A11) and (A12) by $-f_p$ and taking the complex conjugate of each expression.

(4) DC-bus voltage controller

The instantaneous active power flowing into the MMC can be obtained as:

$$P = 1.5\big(e_d i_{gd} + e_q i_{gq}\big) \tag{A14}$$

where $e_d$ and $e_q$ are the grid voltages in $dq$-frame and $i_{gd}$ and $i_{gq}$ are the grid currents in $dq$-frame.

Since the grid current in the $q$-axis is set to zero for the unity power factor, the small-signal model of active power can be simplified as:

$$P_p = 1.5\big(e_{d0} i_{gdp} + e_{dp} i_{gd0}\big) \tag{A15}$$

Expressed in the energy $C_{dc}(v_{dc})^2/2$ stored in the dc capacitor, the dc-link dynamics is given by:

$$0.5 C_{eq} \frac{dv_{dc}^2}{dt} = P - P_L \tag{A16}$$

where $P_L$ is the load power, expressed as $P_L = v_{dc} I_{load}$. $C_{eq}$ represents the equivalent dc capacitance of MMC containing the cable capacitance in dc-bus $C_{DC}$ and equivalent SM capacitance, which is defined as:

$$C_{eq} = C_{DC} + 6 C_{SM} / N \tag{A17}$$

The small-signal model of dc-link voltage is given as:

$$C_{eq} v_{dc0} \frac{dv_{dcp}}{dt} = P_p - P_{Lp} \tag{A18}$$

Substituting (A15) into (A18), we obtain:

$$C_{eq} v_{dc0} s v_{dcp} = 1.5\big(E_1 i_{dp} + I_g u_{dp}\big) - v_{dcp} I_{load}$$
$$\Rightarrow v_{dcp}(s) = \frac{1.5\big(E_1 I_{gp0} + I_g E_{p0}\big)}{I_{load} + C_{eq} v_{dc0} s}$$
$$= G_{dc1}(s) I_{gp0} + G_{dc2}(s) E_{p0}, \qquad f = f_p - f_1 \tag{A19}$$

where:

$$G_{dc1}(s) = \frac{1.5 E_1}{I_{load} + C_{eq} v_{dc0} s}, G_{dc2}(s) = \frac{1.5 I_g}{I_{load} + C_{eq} v_{dc0} s} \tag{A20}$$

The dc-link voltage controller is given as:

$$i_{dref} = H_{dc}(s)\Big[ v_{dcref} - v_{dc} \Big] \tag{A21}$$

where $H_{dc}(s)$ is the transfer function of the dc-link voltage controller.

Considering the dc-link voltage dynamics, the small-signal reference current in the *d*-axis is expressed as:

$$
\begin{aligned}
i_{drefp}(s) &= -H_{dc}(s)v_{dcp} \\
&= -H_{dc}(s)G_{dc1}(s)I_{gp0} - H_{dc}(s)G_{dc2}(s)E_{p0}, f = f_p - f_1
\end{aligned} \tag{A22}
$$

Then, the perturbation modulation index in the d-axis caused by the dc-link voltage dynamics is obtained as:

$$
\begin{aligned}
m_{dp}(s) &= H_i(s)i_{drefp} \\
&= -H_i(s)H_{dc}(s)G_{dc1}(s)I_{gp0} - H_i(s)H_{dc}(s)G_{dc2}(s)E_{p0}, f = f_p - f_1
\end{aligned} \tag{A23}
$$

After transformation back to the *abc* frame, the perturbation modulation voltage at $f_p - f_1$ creates a component at $f_p$ and another at $f_p - 2f_1$, given as:

$$
m_p = \begin{cases}
\begin{aligned}
& -0.5H_i(s - j\omega_1)H_{dc}(s - j\omega_1)G_{dc1}(s - j\omega_1)I_{gp0} \\
& -0.5H_i(s - j\omega_1)H_{dc}(s - j\omega_1)G_{dc2}(s - j\omega_1)E_{p0},
\end{aligned} & f = f_p \\[1em]
\begin{aligned}
& -0.5H_i(s + j\omega_1)H_{dc}(s + j\omega_1)G_{dc1}(s + j\omega_1)I_{gp0} \\
& -0.5H_i(s + j\omega_1)H_{dc}(s + j\omega_1)G_{dc2}(s + j\omega_1)E_{p0},
\end{aligned} & f = f_p - 2f_1
\end{cases} \tag{A24}
$$

Based on the above analysis, the dc-link voltage controller model can be obtained as:

$$
\boldsymbol{m}_{up} = \boldsymbol{T}\boldsymbol{e}_p + \boldsymbol{Q}\boldsymbol{i}_{gp}, \boldsymbol{m}_{lp} = -\boldsymbol{T}\boldsymbol{e}_p - \boldsymbol{Q}\boldsymbol{i}_{gp} \tag{A25}
$$

where $\boldsymbol{T}$ is a $(2n + 1) \times (2n + 1)$ zero matrix except for the $(n + 1, n + 1)$th element and the $(n + 1, n - 1)$th element. The $(n + 1, n + 1)$th element and the $(n + 1, n - 1)$th element are the same, denoted as:

$$
-0.5H_i\big[j(\omega_p - \omega_1)\big]H_{dc}\big[j(\omega_p - \omega_1)\big]G_{dc2}\big[j(\omega_p - \omega_1)\big] \tag{A26}
$$

Similarly, the $(n + 1, n + 1)$th element and the $(n + 1, n\text{-}1)$th element in $\boldsymbol{Q}$ are expressed as:

$$
-0.5H_i\big[j(\omega_p - \omega_1)\big]H_{dc}\big[j(\omega_p - \omega_1)\big]G_{dc1}\big[j(\omega_p - \omega_1)\big] \tag{A27}
$$

For the dc-link voltage controller model under negative-sequence perturbation voltage, the $\boldsymbol{T}$ and $\boldsymbol{Q}$ are the $(2n + 1) \times (2n + 1)$ zero matrix except for the $(n + 1, n + 1)$th element and$(n + 1, n + 3)$th element. The elements can be obtained by replacing $f_p$ in (A26) and (A27) by $-f_p$ and taking the complex conjugate.

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
