# Peer review of "Effect of Frequency Coupling on Stability Analysis of a Grid-Connected Modular Multilevel Converter System"

_energies, doi:10.3390/en14206580_

Round 1

Reviewer 1 Report

The paper discusses the effect of frequency coupling on the stability assessment of modular multilevel converters. The paper is interesting and the results are useful. Some comments about the paper will be listed below according to the page numbers (not to their significance):

  • The term "frequency coupling" is fundamental in the work. However, it's not introduced to the reader in either the abstract or the introduction. This term should be defined so the reader understands the basic idea of the work.
  • Page 1, line 18: "Based on the analysis result, a simple..." this statement is not clear
  • Page 1, line 21: "Finally, the theoretical analysis is validated by the time-domain simulations results". What about the experimental results you have ?
  • Page 2, line 60: "The offshore AC-side MMC impedance model is..." what is the offshore AC-side ? I think it should be just the AC side.
  • Page2, line 62: " It is universally acknowledged that ignoring the frequency coupling affects the stability analysis of the interconnection systems." This sentence should be re-phrased.
  • Page 2, line 63: "A complete MMC impedance model is proposed in [26], and the frequency coupling causes MMC impedance to be affected by grid impedance". This sentence is not clear and should be rephrased. 
  • "It is universally acknowledged that ignoring the frequency", same.
  • Page 3, Fig.2: us and uc should be defined. 
  • Page 3, line 112: "Then the arm consisted of individual SMs can be treated as a averaged model. Not clear and not correct.
  • Page 5, Fig. 3: f1 looks like the fundamental frequency, but not defined.
  • Fig. 8 shows a perfect match  between the theoretical and experimental analysis. However, the grid resistance doesn't appear in this theoretical analysis. We all know that this resistance should exist and it may affect this perfect matching. Can you please explain this issue ?
  • Again in Table 2, page 10, the grid resistance (impedance) doesn't appear.
  • How can the authors assess the stability from the bode plots in Fig. 16 and 19 ? Please give more comments on the phase and gain margins.
  • Are the results in Fig. 18 conducted with the same controller ? please show the structure and values of this controller
  • Please check the caption of this figure (Fig. 18). 

Author Response

Thanks for your constructive comments. The response is in the PDF file.

Reviewer 2 Report

Some of the analysis/discussion of the results could be a bit more critical. For example, the comparative analysis between analytical and experimental validations of the admittance responses was not adequate - it was simply glossed over by saying "they match well" despite some clear differences in the results.

Author Response

(The authors gave the same response as above.)

Reviewer 3 Report

The paper presents an interesting topic but I have some remarks and questions : 

1* The literature part has to be improved and updated.

2* The quality of the presented figures has to be enhanced.

3* How the authors could test the efficiency and the sensitivity of the proposed strategy with the transient variation?

4* a comparative study is important.

5* The limitation of the proposed strategy has to be included in the conclusion. 

Author Response

(The authors gave the same response as above.)

Round 2

Reviewer 3 Report

I would like to thank the authors for their efforts to improve the paper. The paper is now ready to be published in Energies.